# Low-Dose Eribulin Promotes NK Cell-Mediated Therapeutic Efficacy in Bladder Cancer

**DOI:** 10.3390/cancers16223875

**Published:** 2024-11-19

**Authors:** Zaineb Hassouneh, Onika D. V. Noel, Niannian Ji, Michelle E. Kim, Jordan Svatek, Robert S. Svatek, April L. Risinger, Neelam Mukherjee

**Affiliations:** 1Department of Microbiology, Immunology & Molecular Genetics, University of Texas Health San Antonio (UTHSA), San Antonio, TX 78229, USA; hassouneh@livemail.uthscsa.edu; 2Department of Urology, University of Texas Health San Antonio (UTHSA), San Antonio, TX 78229, USA; noelo@uthscsa.edu (O.D.V.N.); jin3@uthscsa.edu (N.J.); kime3@uthscsa.edu (M.E.K.); svatekj@uthscsa.edu (J.S.); svatek@uthscsa.edu (R.S.S.); 3Mays Cancer Center, San Antonio, TX 78229, USA; risingera@uthscsa.edu; 4Long School of Medicine, University of Texas Health San Antonio (UTHSA), San Antonio, TX 78229, USA; 5Department of Pharmacology, University of Texas Health San Antonio (UTHSA), San Antonio, TX 78229, USA

**Keywords:** bladder cancer, NK cells, immunotherapy, innate lymphoid cells

## Abstract

Although bladder cancer (BCa) is known for its immunogenic characteristics, it often shows limited responsiveness to currently approved immunotherapies. Our previous research identified natural killer (NK) cells as significant contributors to improved survival outcomes in BCa patients. In this study, we found that eribulin, a microtubule destabilizer typically used for breast cancer, can activate NK cells. While clinical trials are exploring the use of eribulin in BCa treatment, the underlying mechanism was previously unknown. We demonstrated that low-dose eribulin activates NK cells, leading to reduced tumor burden and improved survival in various murine models. Mechanistically, eribulin enhanced NK cell migration and cytotoxicity against BCa cells, promoted an anti-tumor NK cell phenotype, and reduced exhaustion markers, uncovering an unexpected role of low-dose chemotherapy in boosting NK cell-mediated anti-tumor immunity. Given eribulin’s clinical availability, our findings support its potential as a rapid bench-to-bedside immune-adjuvant treatment for BCa.

## 1. Introduction

Bladder cancer (BCa) is the most common urinary tract malignancy, impacting over 80,000 individuals annually in the U.S. [1,2]. The high tumor mutational burden (TMB) of BCa makes it a perfect candidate for immunotherapies, as evidenced by the FDA approval of all five immune checkpoint inhibitors (ICI) in BCa [3]. However, despite the success of immunotherapies in the past decade, approximately 80% of patients with BCa do not respond to current treatments, leading to a lack of reliable alternatives aside from a radical cystectomy [1]. Research on improving BCa immunotherapy response is mostly focused on T cells, while studies on innate immunity, including natural killer (NK) cells, remain limited. 

NK cells are regulated by various cell surface activation and inhibitory receptors, allowing NK cells to recognize and mediate direct cytotoxicity against tumor cells without harming healthy cells [4]. The capacity of NK cells to mediate cytotoxicity quickly and without prior priming has made them an attractive target for cancer immunotherapy. We previously demonstrated that activated intratumoral NK cells are associated with improved survival in patients suffering from BCa and that higher stages of BCa show an accumulation of dysfunctional NK cells [5]. Similar findings of dysfunctional intratumoral NK cells and their phenotypic shifts have been reported in other malignancies [6,7]. Recently, a study revealed that peripheral circulating NK cells could differentiate into two divergent pathways based on cues in the tumor microenvironment (TME): the CD49a^+^ CD103^+^ (intraepithelial innate lymphoid cell 1; ieILC1-like) subset, which exhibited potent antitumor and immunosurveillance functions, and the NR4A2-expressing CD49a^−^ (intermediate ILC1; intILC1-like) subset, which showed poor antitumor function [8]. 

In our efforts to identify current therapies that could increase the activation of NK cells in BCa, we identified eribulin, a microtubule-destabilizing chemotherapeutic that has unique immunological effects as compared to other types of chemotherapy [9]. Eribulin is currently used in the treatment of locally advanced or metastatic breast cancer, where it has demonstrated a survival advantage likely due in part to its ability to stabilize tumor vasculature [10] and enhance antitumor immune responses [11,12,13]. We found that eribulin can directly induce STING-dependent and -independent interferon production in monocytes in a manner distinct from microtubule-stabilizing taxanes to enhance antitumor immunity and promote tumor regression in breast cancer models [14,15]. The ability of eribulin to enhance antitumor immunity in breast cancer, combined with the fact that it is currently being tested in clinical trials for its integration into the existing standard of care for metastatic BCa [16], prompted us to determine whether eribulin could increase NK cell activation and enhance the intratumoral infiltration of NK cells in BCa. To the best of our knowledge, no studies have investigated the potential immunological effects of low-dose eribulin in primary BCa to provide a rationale for the use of this highly effective but mechanistically underappreciated chemotherapeutic to enhance immunogenicity in BCa. In this study, we tested the direct effects of eribulin on NK cell activation and BCa-directed migration in vitro, utilizing both cell lines and primary BCa patient-derived NK cells. Furthermore, we demonstrate that the in vivo efficacy of eribulin in both syngeneic and humanized BCa models, including localized treatment via intravesical injection, is NK cell-dependent and is associated with enhanced NK cell activation. These data provide a mechanistic rationale for the use of eribulin in BCa treatment regimens, including the potential for combination with existing immunotherapy.

## 2. Materials and Methods

### 2.1. Cell Lines

NK92 and BCa cell lines were maintained at a seeding concentration of approximately 0.1 × 10^6^ cells/mL for 2–3 days at 37 °C with 5% CO_2_ (not exceeding 0.8 × 10^6^ cells/mL). The NK92 cell line (donated by Dr. Emily Macy, Columbia University) was cultured in complete MEMα (cMEMα) media containing Minimum Essential Medium-alpha (Gibco, Grand Island, NY, USA) supplemented with 12.5% heat-inactivated horse serum (Gibco), 12.5% heat-inactivated fetal bovine serum (FBS; Gibco), 0.2 mM myoinositol (ThermoFisher Scientific, Waltham, MA, USA), 0.02 mM folic acid (ThermoFisher Scientific), 0.1 mM β-mercaptoethanol (Gibco), 100 U/mL IL-2 (Millipore, Burlington, MA, USA), 100 U/mL penicillin (Gibco), 100 μg/mL streptomycin (Gibco), and 2 mM fresh L-glutamine (Corning, Corning, NY, USA). T24 (donated by Dr. Rita Ghosh at University of Texas Health San Antonio) and UM-UC3 (MD Anderson) human BCa cell lines and MB49 mouse BCa cell line (donated by Dr. Jeffrey Schlom at NIH) were maintained in Dulbecco’s modified Eagle’s medium (DMEM; ThermoFisher Scientific) supplemented with 100 U/mL penicillin, 100 μg/mL streptomycin, 10% FBS, 1% Non-Essential Amino Acid solution (ThermoFisher Scientific), and 2 mM fresh L-glutamine. H5100 media (StemCell Technologies, Vancouver, BC, Canada) supplemented with 10 nM hydrocortisone (StemCell Technologies) was used to maintain the NK92.MI cell line (ATCC, Manassas, VA, USA). All cell lines were validated by STR profiling and routinely evaluated for mycoplasma contamination.

### 2.2. Patient-Derived NK Cells

Blood and tissue samples were collected from BCa patients recruited through an Institutional Review Board (IRB)-approved observational cohort study (#BCR20120159H). All patients were 18 years of age or older with confirmed or suspected BCa diagnoses and provided written consent. Patient demographics and medical records were entered into a secure web-based REDCap database. This study’s involvement with human subjects complies with the Declaration of Helsinki. Tumor tissue samples were surgically excised through transurethral resection of the bladder tumor (TURBT) or cystectomy and maintained in Roswell Park Memorial Institute (RPMI) 1640 medium supplemented with 100 U/mL penicillin and 100 μg/mL streptomycin and transported on ice. Samples were digested as previously described [5]. Briefly, fresh tumor tissues were washed with Dulbecco’s phosphate-buffered saline (DPBS; Gibco), minced into 1–2 mm pieces, and incubated in a digestion solution. Following digestion, samples were washed with complete RPMI media containing 10% fetal bovine serum, and the single-cell suspensions (SCS) were cryopreserved at −150 °C until later use. Peripheral blood mononuclear cells (PBMCs) were obtained from BCa patient blood samples by Ficoll–Paque (GE Healthcare, Chicago, IL, USA) centrifugation and cryopreserved at −150 °C until later use. Prior to immune assay and analysis, PBMCs or tumor samples were thawed using Anti-Aggregate Wash (C.T.L., Cellular Technology Limited, Cleaveland, OH, USA) for live cell count by Countess Cell Counter (Invitrogen, Waltham, MA, USA) using 0.4% Trypan Blue Solution (Invitrogen).

### 2.3. Flow Cytometry

NK92.MI cells were collected by centrifugation (300 g/rcf for 7 min at room temperature) and resuspended in cold flow buffer (2% FBS in DPBS; FB) at a concentration of 2 × 10^6^ cells/mL then plated in a 96-well U-bottom plate (1 × 10^5^ cells/well). Cells were then treated with 100 nM eribulin (EISAI Inc., Woodcliff Lake, NJ, USA) and incubated for 3 h at 37 °C with 5% CO_2_ prior to washing with 150 μL of FB. Cells were then incubated in 50 µL of FB containing Hu-TruStain FcX (2 µL/test; BioLegend, San Diego, CA, USA) at 4 °C for 10 min, then another 50 µL of FB containing a master mix of fluorochrome-conjugated antibodies (0.1 µg/test; Table 1) and 2× of fixable viability dye (FVD)-eFluor 455UV (1000× stock; Invitrogen) at 4 °C for 45 min. Cells were then washed with 150 µL of cold FB before fixation in 2% paraformaldehyde (PFA). For intracellular staining, cells were pretreated with GolgiPlug (BD Biosciences, Franklin Lakes, NJ, USA) in cMEMα media at 37 °C for 4 h prior to staining. Following fixation after cell surface staining, cells were permeabilized using Cytoperm/Cytofix buffer (BD Biosciences) at 4° for 20 min, then washed with 1× permeabilization buffer (BD Biosciences) and stained with 100 µL of permeabilization buffer containing intracellular antibody master mix (0.2 µg/test; Table 1) at 4 °C for 45 min. Samples were analyzed on an LSRII flow cytometer (BD Biosciences) and using the FlowJo software V10.10. For eribulin treatment in patient-derived PBMCs and tumor SCS, cryopreserved patient samples were thawed in 1× C.T.L in RPMI and centrifugated at 300 g/rcf for 7 min at room temperature. Samples were then resuspended in complete RPMI supplemented with 100 U/mL penicillin, 100 μg/mL streptomycin, and 2 mM L-glutamine (cR-10) at 20 × 10^6^ cells/mL and plated in a 96-well U-bottom plate (1 × 10^6^ cells/well) and treated with 100 nM eribulin for 3 h at 37 °C, 5% CO_2_. Following incubation, the samples were stained and fixed as previously described.

### 2.4. Cytotoxicity Assay

T24 BCa cells were trypsinized by 0.25% Trypsin/EDTA (Corning) for ~1 min, and digestion was stopped by 2× volume of cDMEM-10 media. The cells were then centrifugated, and the pellet was resuspended in 0.1% bovine serum albumin (BSA, Fisher Scientific, Hampton, NH, USA) at 10 × 10^6^ cells/mL. Carboxyfluorescein succinimidyl ester (CFSE, Life Technologies, Carlsbad, CA, USA) was added so the final concentration was 2 µM, and cells were incubated at 37 °C for 20 min. Following incubation, T24 cells were washed with serum-free MEMα media, spun down, and resuspended at 8 × 10^5^ cells/mL in cMEMα media. Cells were incubated for 30 min before the assay to purge CFSE. Meanwhile, NK92 cells were collected by centrifugation and resuspended at 4 × 10^6^ cells/mL in cMEMα media. CFSE-labeled target cells were mixed thoroughly with large-orifice tips and incubated with NK92 cells at an effector:target (E:T) ratio of 5:1 (2 × 10^5^ NK cells and 4 × 10^4^ T24 BCa cells per well) for 4 h in a 96-well plate. After incubation, cells were washed and stained with FVD-eFluor455UV and fixed with 2% PFA. Analysis was performed using the LSRII flow cytometer (BD Biosciences) and the FlowJo software. Cytotoxicity was defined as the percentage of T24 BCa cells killed (%FVD^+^ cells from total CFSE^+^ cells).

### 2.5. Migration Assay

T24 BCa cells were digested and resuspended at 0.4 × 10^6^ cells/mL in cDMEM-10 media. In a 24-well plate, 0.2 × 10^6^ T24 BCa cells were plated and incubated overnight at 37 °C, 5% CO_2_. Prior to the migration assay, NK92 cells were collected by centrifugation and resuspended in FB, and then pre-stained with PE-conjugated anti-CD45 antibody (clone HI30, BioLegend). NK92 cells were then washed and resuspended in serum-free MEMα media at 5 × 10^6^ cells/mL. T24 BCa cells were rinsed with serum-free MEMα media to remove non-adherent cells, and 600 µL of serum-free MEMα media was added to each well, with eribulin (100 nM) or vehicle control (DMSO). The trans-well membrane was then wet and loaded with 100 µL of NK92 cells and placed into the well. The 24-well plate was incubated at 37 °C, 5% CO_2_ for 4 h, and then migrated cells were collected; following this, wells were washed with DPBS and pooled into a 1.5 mL tube. The cell suspension was spun down and resuspended in 100 µL of FB before staining with FVD and fixation in 2% PFA. NK cell migration was determined by flow cytometry using the LSRII flow cytometer.

### 2.6. RNA Sequencing

NK92 cells were resuspended in cMEMα at 1 × 10^6^ cells/mL and pretreated for 3 h with 100 nM eribulin or DMSO vehicle as a control. Following treatments, RNA was extracted using the RNeasy Mini Kit (Qiagen, Cat. No. 74104 and 74106) according to the manufacturer’s suggested protocol. RNA samples were flash-frozen and kept at −80 °C until the time of shipment. RNA sequencing and analysis were performed by Novogene Corporation Inc (Beijing, China). Sequencing and construction of cDNA was performed using the Illumina NovaSeq 2000 utilizing a short-read strategy. Sample quality was guaranteed (>85% bases with Q30 or higher). Gene expression was quantified, and the fold change following eribulin treatment compared to vehicle treatment was calculated. 

### 2.7. Orthotopic Model of Intravesical Eribulin Treatment

C57BL/6 female mice aged 7–9 weeks (Jackson Laboratory, Bar Harbor, ME, USA) were used to establish an orthotopic BCa model. Mice were anesthetized using an isoflurane/O_2_ chamber prior to catheter insertion. After ensuring the bladder was completely voided, 100 μL of poly-L-lysine (Sigma-Aldrich, St. Louis, MO, USA) was instilled intravesically using a 24 G/0.56-inch catheter (Becton Dickinson, Franklin Lakes, NJ, USA) for a minimum of 30 min. Mice were then challenged with 8 × 10^5^ MB49 BCa cells (100 µL of MB49 murine BCa cells resuspended in DPBS at 1.6 × 10^6^ cells/mL to account for dead volume), after ensuring that all poly-L-lysine was drained, for one hour. Mice were treated once a week by intravesical instillation of 100 μL of eribulin (1.8 mg/kg) or DPBS for 4 weeks, beginning on day one post-challenge. For NK cell depletion, mice were injected intraperitoneally with 50 μL of anti-Asialo GM-1 (FUJIFILM Wako Chemicals, Richmond, VA, USA) 4 days before the first treatment and weekly thereafter. Mice were euthanized, and bladders were excised and weighed prior to digestion. Bladder tumors were minced into 1–2 mm pieces and suspended in 3 mL of RPMI with 1.5 mg/mL collagenase IV (Sigma-Aldrich) and 0.25 mg/mL of DNase I (Sigma-Aldrich) and incubated at 37 °C, 5% CO_2_ for 45 min with agitation [17]. Digestion was stopped by the addition of 7 mL of cR-10 and passed through a 100 μm strainer. The live cell counts and viability for all single-cell suspensions were then recorded prior to centrifugation. Samples were resuspended at 10 × 10^6^ cells/mL in cR-10 and plated in a 96-well U-bottom plate at 100 μL/well for flow staining. For intracellular staining, samples were pre-treated with 1× Cell Activation Cocktail with Brefeldin A (BioLegend) for 5 h, then stained as described above.

### 2.8. Adoptive Transfer in a Humanized Murine Model

Female NSG mice used for the humanized model were purchased from Jackson Laboratories (Strain #:005557). Both UMUC-3 BCa cells and NK92.MI cells were cultured in respective media until a sufficient number of cells could be harvested. Cells were harvested, spun down, and resuspended in DPBS at an effector:target (E:T) ratio of 3:1. Mice were then injected subcutaneously with a mix of UMUC-3 and NK92.MI cells, with a total of 3 × 10^6^ NK92.MI and 1 × 10^6^ UMUC-3 cells in either flank. Tumor volume (TV) was measured three times a week and calculated according to the following formula:$$\mathrm{TV}\left({\mathrm{mm}}^{3}\right)=\mathrm{L(mm)}\times \mathrm{W(mm)}\times (\frac{\mathrm{W(mm)}}{2})$$

Once TV reached approximately 50 mm^3^, eribulin (1 mg/kg) was administered intratumorally once a week. 

### 2.9. Statistical Analysis

Statistical analysis was performed using a two-sided, unpaired *t*-test for comparison of protein expression, migration, and cytotoxicity. A two-sided, paired Student’s *t*-test was used in human samples treated with eribulin. In cases where results did not pass a normality test, such as bladder weights, a Mann–Whitney test was used. A two-way ANOVA was used to compare tumor volume curves. Kaplan–Meier survival curves were generated for tumor curves, where 1 was defined as when any mouse-bearing tumor(s) ≥ 1500 mm^3^. Survival was then calculated using a Log-Rank (Mantel–Cox) test. *p*-values below 0.05 were considered significant. For bulk RNA sequencing, statistical and differential gene analyses were performed using the edgeR v 3.22.5 in R, and heatmaps were generated using the R package Heatmap.

## 3. Results

### 3.1. Low-Dose Eribulin Exerted Anti-Tumor Effects in Multiple Murine Models of Bladder Cancer via NK Cells

To assess the NK cell-mediated anti-tumor effects of eribulin treatment in BCa, we developed a syngeneic and orthotopic BCa model where C57BL/6 mice were challenged by intravesical instillation of MB49 murine BCa cells and administered intravesical eribulin (1.8 mg/kg) once a week. Typically administered systemically, eribulin’s localized treatment in the bladder has not been previously explored [18]. Localized intravesical eribulin treatment significantly reduced tumor burden as compared to vehicle control (Figure 1A). Eribulin treatment was found to increase the population of bladder-tumor-infiltrating NK cells (Figure 1B) as well as markers of their activation (NKG2D), degranulation (CD107a), and tissue residency (CD49a and CD103) compared to vehicle-treated control mice (Figure 1C–F) [4,8,19]. Eribulin treatment also significantly reduced the expression of PD-1 and CD73 on NK cells (Figure 1G, Appendix A), both of which are markers associated with immune-suppressive or exhausted NK cells [20,21]. To further confirm the role of NK cells in eribulin-mediated anti-tumor effects, we carried out eribulin treatment following NK cell depletion in our orthotopic BCa model. Strikingly, the antitumor efficacy of eribulin was abolished upon depletion of NK cells, further supporting the NK-dependent efficacy of eribulin (Appendix A). Lastly, the NK cell-dependent anti-tumor effects of eribulin were also confirmed in our humanized BCa/NK cell adoptive transfer model where intratumoral weekly injections of 1 mg/kg eribulin significantly reduced tumor volume (Appendix A) and improved the survival of mice compared to vehicle-treated control mice (Appendix A).

### 3.2. Low-Dose Eribulin Mediated NK Cell Activation, Migration, and Cytotoxicity Against Bladder Cancer Cells In Vitro

To test the direct effects of eribulin on NK cells in vitro that could underlie the NK-mediated antitumor efficacy in vivo, NK92.MI cells were treated with 100 nM eribulin for 3 h, a clinically relevant serum concentration that is achieved with systemic dosing [22]. Eribulin treatment enhanced NK cell cytotoxicity against human T24 BCa cells (Figure 2A) but was not sufficient to induce the death of BCa cells on its own (Figure 2B), again confirming the involvement of NK cells in its treatment effects. RNA sequencing of eribulin-treated NK cells showed that transcripts of cytotoxicity-related genes were upregulated following eribulin treatment, specifically genes encoding perforin and TNFα (Figure 2C,D, Appendix A) along with genes involving adhesion, secreted cytokines, cell cycle, and quiescence (Appendix A). Consistent with our in vivo data, in vitro flow cytometry data show that eribulin-treated NK92.MI cells have increased expression of the activation receptors NKp30 and NKG2D as well as the degranulation marker CD107a (Figure 3A–C). Eribulin treatment also increased the BCa-directed migration of NK cells in vitro (Figure 3D) and increased the transcription of *CCR1* and *CCR7*, chemokines involved in NK cell migration (Figure 3E,F, Appendix A) [23,24].

### 3.3. Low-Dose Eribulin Promoted the Shift of Patient-Derived Intratumoral NK Cells into Anti-Tumor ieILC1-like Cells 

Depending on environmental cues, NK cells have been shown to change their phenotype, ranging from cytotoxic activated to exhausted phenotypes [25]. In head and neck cancer, different subsets of NK cells were recently identified based on levels of cytotoxicity and surface expression of different markers. Among them, two predominant subsets were identified: the cytotoxic CD49a^+^ CD103^+^ subset found to resemble cytotoxic ieILC1, termed NK1, and the pro-tumorigenic, dysfunctional CD49a^−^NR4A2^+^ subset, found to resemble intILC1, termed NK2 [8]. We first sought to determine if similar NK cell phenotypes were observed among bladder-tumor-infiltrating NK cells as compared to peripheral NK cells by characterizing patient-derived tumor and peripheral blood NK cells using these criteria. We found that, compared to the peripheral blood, the ieILC1-like NK1 population was highly enriched in bladder tumors (Appendix A). Further phenotypic analysis highlights the functional differences between the NK1 and NK2 subsets in the bladder cancer microenvironment. The NK1 subset’s higher expression of activation receptors such as NKG2D and NKp30, coupled with lower levels of the exhaustion marker CD73 (Figure 4A–C), suggests that NK1 cells are less prone to dysfunction and maintain greater cytotoxic potential. This distinction could be crucial in understanding their respective roles in anti-tumor immunity and may offer insights into optimizing NK cell-targeted therapies by selectively enhancing the more functionally competent NK1 subset. Interestingly, eribulin treatment significantly skewed the phenotypic expression of tumor-infiltrating NK cells towards the NK1 phenotype and decreased the NK2-cell-like population (Figure 5A,B). This was also validated by the decreased expression of NK2-associated genes including *NR4A2*, *NR4A3*, *CTLA4*, and *TIGIT*, as well as the increased expression of NK1-associated genes, including *EOMES*, *IFIT3*, *MKI67*, and *TBX21*, upon eribulin treatment in the NK92 cell line (Figure 5C–J) [8]. These results underscore the therapeutic potential of eribulin in modulating NK cell phenotypes within the TME, presenting a promising strategy to enhance anti-tumor responses by promoting the NK1 subset while reducing the prevalence of dysfunctional NK2 cells.

## 4. Discussion

We have previously shown the role of NK cells in the prognosis of BCa; however many of the current immunotherapies mainly target adaptive CD8 T-lymphocytes, and new approaches are needed to enhance NK cell-mediated antitumor efficacy [5,26]. Microtubule-targeted chemotherapeutics, including eribulin, remain key treatments for solid tumors and hematological malignancies, primarily due to their antimitotic effects, which are frequently linked to immunosuppressive outcomes [27]. However, eribulin has been shown to promote antitumor immunity in cellular, animal, and human studies, including enhancing the efficacy of anti-PD-1 therapy [14,15,28]. Eribulin has also been shown to increase the intratumoral infiltration of NK cells in a breast cancer orthotopic model [12], and the specific depletion of NK cells in xenograft models of non-small cell lung cancer and melanoma was found to decrease the anti-tumor effects of eribulin [29]. Additionally, eribulin is being currently tested in combination with ICIs, such as avelumab (NCT03502681), or other chemotherapy drugs, such as gemcitabine (NCT04579224), in metastatic or refractory BCa [16,30]. This study elucidates eribulin’s impact on NK cell functionality in BCa, utilizing both in vitro and in vivo models, including primary patient-derived tumor tissue. We also provide the first mechanistic rationale for leveraging eribulin as an immune adjuvant to boost NK cell-mediated antitumor efficacy in BCa.

Examination of the BCa TME revealed increased infiltration of activated and cytotoxic NK cells in the eribulin-treated group compared to controls. This eribulin-mediated NK cell activation was also confirmed in vitro, demonstrated by the increased BCa cell death in the presence of NK cells treated with eribulin, as well as the increased expression of activation markers on eribulin-treated NK cells. This finding is supported by previous studies that showed eribulin treatment enhanced tumor vascularization via intussusceptive angiogenesis, a phenomenon speculated to heighten intratumoral immune infiltration [10,31]. In addition to activating NK cells, low-dose eribulin also enhanced their migration towards BCa cells, as indicated by the transcriptional upregulation of tumor-homing chemokine receptors. Together, our data demonstrate that the anti-tumor effect of eribulin in BCa is NK cell-specific and independent of vascular restructuring.

Low-dose eribulin also increased the population of intratumoral NK cells that express tissue residency markers such as CD49a and CD103, which further supported the NK cell-specific effect of eribulin. This is particularly interesting since CD49a, or integrin alpha-1, is a marker specific to ILC1s, while NK cells generally express integrin alpha-2, or CD49b [32,33]. Recently CD49a and C103 co-expressing NK cells have also been referred to as tissue-resident (tr) NK cells [34]. Eribulin treatment of bladder tumors induced the differentiation of tumor-infiltrating NK cells into a distinct subset characterized by high cytotoxicity and the expression of CD49a and CD103. This finding aligns with a previous study on head and neck squamous cell carcinoma, which identified a similar NK cell subset, termed NK1, and likened it to the intraepithelial ILC1 (ieILC1) subset [8]. These NK1 cells showed an increase in cytotoxic and proliferative capacity and a decrease in the immunosuppressive marker, NR4A2 [8]. We found that eribulin treatment also decreased the transcripts of NR4A2 in NK cells, possibly inducing differentiation of these NK cells into ieILC1-like NK1 cells. 

While the exact mechanism by which NK cells differentiate into NK1 vs. NK2 cells is unclear, Moreno-Nieves et al. show a possible synergy between IL-15 and TGFβ, as well as direct NK–cancer cell contact, which is required for NK1 differentiation [8]. NK-ILC1 plasticity remains to be fully elucidated; however, mechanistic studies have found that TGFβ imprinting is specifically involved in NK cell transdifferentiation [35]. Inhibition of SMAD4 has also been previously shown to induce the differentiation of NK cells to ILC1 through non-canonical TGFβ signaling, resulting in increased CD49a expression [35]. Eribulin has also been found to affect TGFβ signaling, specifically through the inhibition of SMAD2/3, resulting in the downregulation of *SNAI1* transcription [36,37]. Our RNAseq data showed that eribulin treatment had a significant effect on TGFβ signaling genes, including the upregulation of *SMURF1*, *RAB35*, *RAB40C*, *RAP2B*, *RHOH*, *RRAGA*, *ARHGEF17*, *ARHGEF2*, and *RAPGEF6*, as well as the downregulation of *SNAI1* (Appendix A), suggesting that eribulin may induce NK cell transdifferentiation to an NK1 phenotype through the suppression of canonical TGFβ signaling. In addition, eribulin has been shown to increase the population of CD103^+^ T lymphocytes and subsequent activation by upregulating the expression of E-cadherin on cancer cells in a murine model of lung cancer, melanoma, and bladder cancer [38]. This may be an additional mechanism by which eribulin increases NK cell recruitment and activation.

BCa is characterized by its high tumor mutational burden, which makes it an attractive target for immunotherapy, including the current gold standard treatment, BCG, and immune-checkpoint inhibitors (ICI), such as pembrolizumab [39]. However, BCG failure is seen in a third of patients, and recurrence following ICI therapy is seen in nearly 80% of patients [39,40]. Consequently, efforts are being taken to alleviate BCG failure in BCa as a result of BCG intolerance or refractory/unresponsive BCa. Various studies are investigating combinations of chemotherapies and immunotherapies, as well as alternative drug delivery, to improve efficacy [40]. Recently, a phase III trial, Checkpoint 901, investigating the combined effects of nivolumab and gemcitabine-cisplatin, showed an increase in the overall survival in patients with unresectable or metastatic BCa [41]. We have demonstrated that selective delipidation of BCG, as a strategy to reduce its immunogenicity and associated pathology, retains its efficacy in reducing tumor burden in vivo and activating innate immune cells in vitro [17]. Clinical studies have also explored combining immunomodulatory drugs with approved BCa treatments, such as ALT-803 and BCG, in high-grade NMIBC [40,42]. The development of alternative therapies, as well as modified administration of current treatments, are also being studied in BCG-unresponsive BCa, including nanoparticle encapsulation and antibody drug conjugates for targeted treatment delivery [40,43]. The ability of eribulin to increase tumor-infiltrating NK cells, particularly the ieILC1-like subset, not only provides a mechanistic rationale for its efficacy but also supports further investigation of eribulin as an intravesical immunomodulating adjuvant in combination with other immunotherapies for BCa treatment, specifically in high-risk NMIBC in patients unresponsive or intolerant to BCG treatment.

This study has important limitations that need to be considered. In vitro cytotoxicity and migration experiments were conducted using a single BCa cell line, which may limit the broader applicability of the conclusions. Nonetheless, our findings are strongly supported by consistent results from in vivo experiments and analyses of patient-derived tumor samples. The number of human samples tested is modest, and while the data are consistent, variability in tumor infiltrates between patients must be acknowledged. Additionally, while our findings from human samples closely mirror data obtained from our orthotopic murine model—repeated with a satisfactory sample size to support our ex vivo data—studies directly assessing the effects of eribulin on NK cells are still emerging. The mechanisms underlying eribulin-mediated activation and migration of NK cells remain to be fully investigated, and the process of NK cell transdifferentiation has not yet been completely elucidated. Despite these limitations, our data demonstrate a consistent and reproducible effect of eribulin on NK cell cytotoxicity. Our proposed mechanism takes into account existing knowledge of eribulin’s immunomodulatory effects and NK cell transdifferentiation, particularly within the tumor. 

## 5. Conclusions

NK cells play a significant antitumor role in BCa and are prognostically relevant, making them an attractive target for immunotherapy in BCa. The current failure of FDA-approved ICI in BCa, however, poses an issue in constructing new ICI therapies specifically targeting NK cells. Our results illustrate the capacity of low-dose and localized eribulin administration to reduce tumor burden in an NK cell-dependent manner in vivo. We further demonstrate the immunomodulatory effects of eribulin treatment on NK cells in vitro. Finally, we provide a mechanistic rationale behind the augmented NK cell functionality following eribulin treatment, providing evidence of an overall phenotypic shift in intratumoral NK cells towards a cytotoxic and tissue-resident phenotype. 

## Figures and Tables

**Figure 1 cancers-16-03875-f001:**
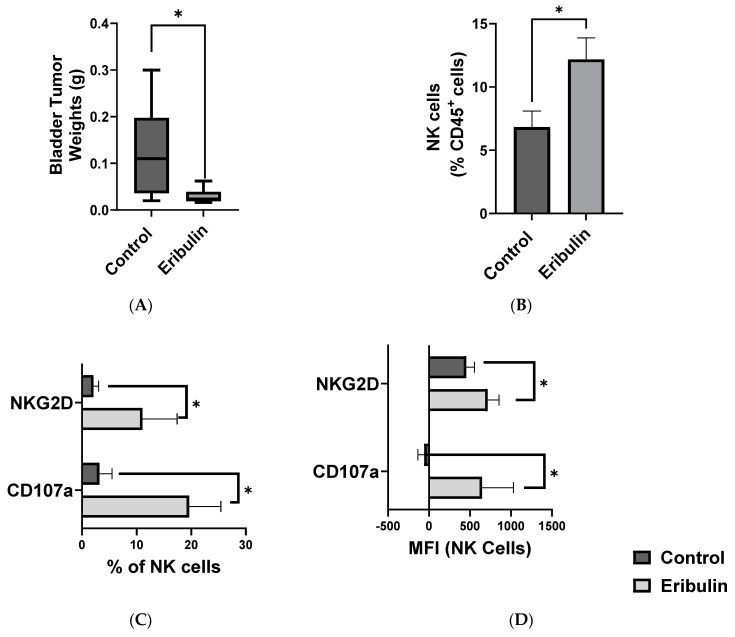
Low-dose eribulin treatment showed NK cell-mediated anti-tumor effects in bladder cancer. C57BL/6 mice were challenged by intravesical instillation of MB49 murine BCa cells and treated with 1.8 mg/kg of eribulin or 100 µL of DPBS as control by weekly intravesical instillation for four weeks beginning on day 1. Shown are the (**A**) bladder weights on d20 pooled from two independent orthotopic experiments (*n* = 10 per group). The bladders were digested and stained for analysis by flow cytometry. (**B**) Illustrated is the proportion of tumor-infiltrating NK cells from total immune infiltrates from one representative experiment. NK cell activation is illustrated as (**C**) the proportion of tumor-infiltrating NK cells expressing NKG2D and CD107a as well as (**D**) their level of expression by mean fluorescent intensity (MFI). NK cell tissue residency is illustrated as (**E**) the proportion of tumor-infiltrating NK cells expressing CD49a and CD103 and (**F**) their level of expression by MFI, whereas NK cell exhaustion is illustrated as the (**G**) proportion of tumor-infiltrating NK cells expressing CD73 and PD-1 from one representative experiment (*n* = 5 per group). MFI: mean fluorescent intensity. * *p* < 0.05 by two-tailed, unpaired *t*-test. ns, not significant.

**Figure 2 cancers-16-03875-f002:**
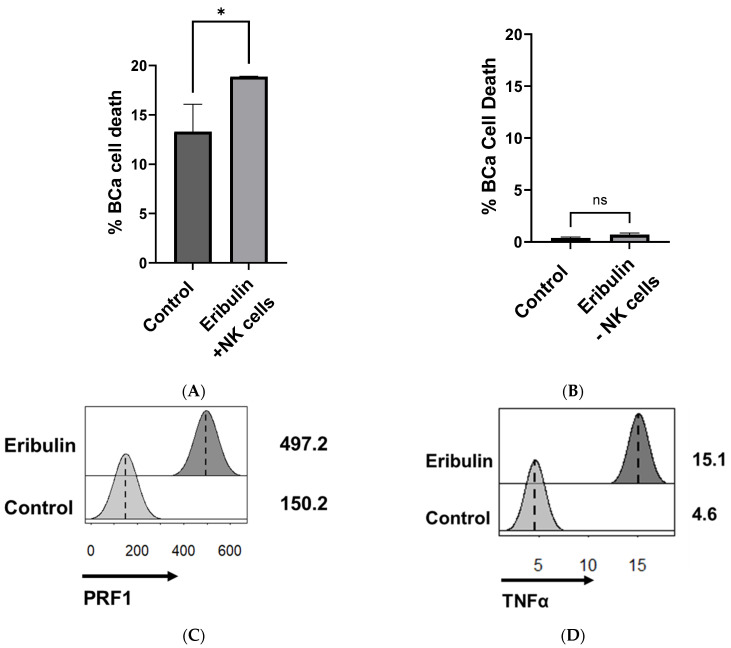
Eribulin treatment increased the cytotoxicity of NK cells against bladder cancer in vitro. T24 BCa cells were pre-stained with CFSE and co-incubated with NK92 cells at an effector:target (E:T) ratio of 5:1 with or without 100 nM eribulin for 4 h. The percentage of BCa cell death, calculated as the percent of FVD^+^ cells out of total CFSE^+^ cells, (**A**) with or (**B**) without the presence of NK cells versus eribulin. The values shown are based on triplicates from representative results of two independent experiments. * *p* < 0.05 by two-sided, unpaired *t*-test. ns; not significant. For bulk RNA sequencing, NK92 cells (1 × 10^6^ cells/mL) were treated with 100 nM of eribulin for 3 h before RNA extraction. Shown are the fragments per kilobase of transcript per million mapped reads (FPKM) of (**C**) *PRF1* and (**D**) *TNFα* illustrated as histograms. Differential analysis of illustrated genes was found to be significant, with *p*-values < 0.05. ns, not significant.

**Figure 3 cancers-16-03875-f003:**
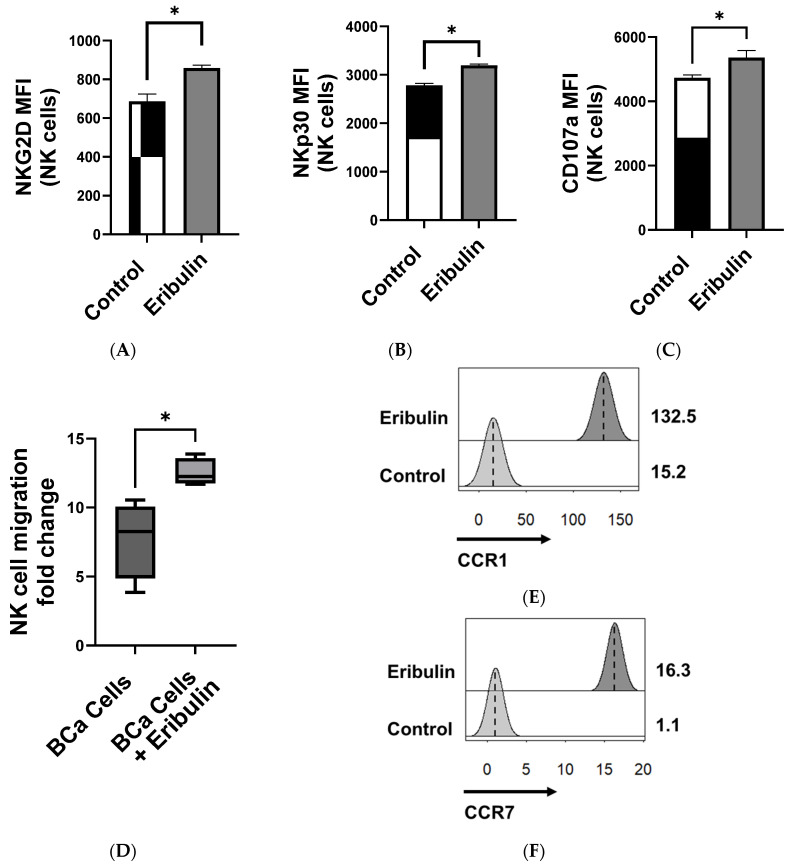
Eribulin increased the activation and migration of NK cells toward bladder cancer in vitro. NK92.MI cells (1 × 10^6^ cells/mL) were treated with 100 nM of eribulin for 3 h prior to staining and flow analysis. Shown is the expression (MFI) of (**A**) NKG2D, (**B**) NKp30, and (**C**) CD107a on live NK cells. Migration of NK92 cells treated with 100 nM of eribulin in response to T24 BCa cells was shown as (**D**) the fold-change of migrated NK cells to BCa cells compared to NK cell migration in serum-free media after 3 h. The values shown are based on triplicates from representative results of two independent experiments. * *p* < 0.05 by two-sided, unpaired *t*-test. Shown are the FPKM of (**E**) CCR1 and (**F**) CCR7 illustrated as histograms. Differential analysis of illustrated genes was found to be significant, with *p*-values < 0.05.

**Figure 4 cancers-16-03875-f004:**
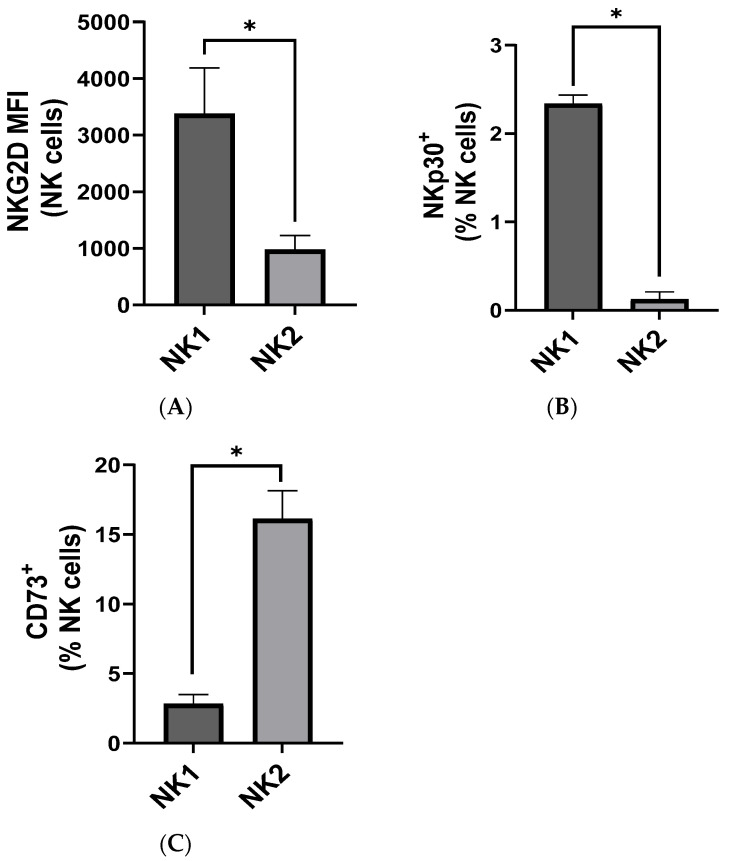
ieILC1-like anti-tumor NK1 cells were more cytotoxic than intILC1 NK2 cells. Patient-derived tumor samples were digested and stained for immune analysis of the tumor-infiltrating NK cells by flow cytometry. NK subsets were defined as NK1 (CD49a^+^ CD103^+^) and NK2 (CD49a^−^). The expression of activation and exhaustion markers on live NK1 and NK2 cells are illustrated as the (**A**) MFI of NKG2D and the proportion of the NK cell population expressing (**B**) NKp30 and (**C**) CD73. Values based on samples derived from three patients. * *p* < 0.05 by two-tailed, unpaired *t*-test.

**Figure 5 cancers-16-03875-f005:**
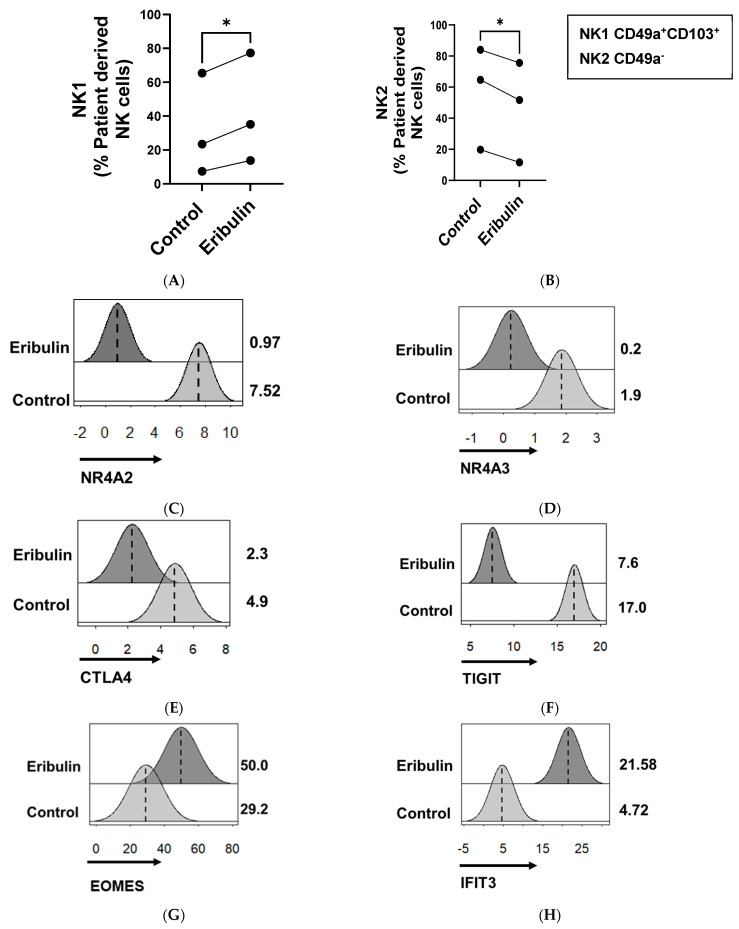
Eribulin treatment shifted patient-derived intratumoral NK cells towards an ieILC1-like anti-tumor NK1 phenotype. Digested human BCa tumor samples were treated with 100 nM of eribulin for 3 h, then stained and analyzed by flow cytometry. The populations of CD49a^+^ CD103^+^ NK cells, termed NK1, and CD49a^−^ NK cells, termed NK2, were analyzed from total live NK cells. The proportion of live patient-derived NK cells differentiated to (**A**) NK1 and (**B**) NK2 of matched patient samples is shown. Values are based on three separate patient samples. * *p* < 0.05 by two-sided, paired Student’s *t*-test. The transcriptional effect of eribulin treatment is illustrated by histograms showing the FPKM of RNA transcripts of NK2-related genes including (**C**) *NR4A2*; (**D**) *NR4A3*; (**E**) *CTLA4*; and (**F**) *TIGIT*- and NK1-related genes including (**G**) *EOMES*, (**H**) *IFIT3*, (**I**) *MKI67*, and (**J**) *TBX21.* Differential analysis of illustrated genes was found to be significant, with *p*-values < 0.05.

**Table 1 cancers-16-03875-t001:** Antibody clones used for flow analysis.

Antibody	Clone (Human)	Clone (Mouse)
CD45	HI30	30-F11
CD3	HIT3a	17A2
CD56	5.1H11	N/A
NKG2D	1D11	CX5
NKp30	P30-15	N/A
NKp46	9E2	29A1.4
CD107a	H4A3	1D4B
CD73	AD2	TY/11.8
Granzyme B	QA16A02	QA16A02
Perforin	dG9	S16009A
IFNγ	4S.B3	XMG1.2
PD-1	EH12.1	29F.1A12
CD103	Ber-ACT8	2E7
CD49a	1090A	HMα1
NK1.1	N/A	PK136

NKG2D: natural killer group 2, member D; IFNγ: interferon gamma; PD-1: programmed cell death protein 1.

## Data Availability

Data are available on request.

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
