# Peer review of "Low-Dose Eribulin Promotes NK Cell-Mediated Therapeutic Efficacy in Bladder Cancer"

_cancers, 2024, doi:10.3390/cancers16223875_

Round 1
Reviewer 1 Report
Comments and Suggestions for Authors
I read this manuscript with interest.
The effort of the authors is timely appropriate and to be commended.
BCa is known for its immunogenic characteristics. From the oncological point, BCG instillations among intermediate and high risk NMIBC represent the gold standard treatment. However, risk stratification for the development of BCG refractory and/or unresponsive disease is lacking and novel emerging molecular alternatives, both considering diagnosis and treatment, are moving to bench to bedside. The authors should discuss these aspects in the Discussion section of the manuscript concerning this topic. Authors should refer to recent review on the topic (doi: 10.3390/pharmaceutics16091154; doi: 10.3390/ijms241612596; doi: 10.1016/j.eururo.2024.08.001; doi: 10.1016/j.euo.2024.05.012).
Immunogenic characteristics could be associated to molecular profiling of BC (subtypes)? E.g. epithelial to mesenchymal transition may enhance these in the tumor microenvironment.
Author Response
Reviewer 1
BCa is known for its immunogenic characteristics. From the oncological point, BCG instillations among intermediate and high risk NMIBC represent the gold standard treatment. However, risk stratification for the development of BCG refractory and/or unresponsive disease is lacking and novel emerging molecular alternatives, both considering diagnosis and treatment, are moving to bench to bedside. The authors should discuss these aspects in the Discussion section of the manuscript concerning this topic. Authors should refer to recent review on the topic (doi: 10.3390/pharmaceutics16091154; doi: 10.3390/ijms241612596; doi: 10.1016/j.eururo.2024.08.001; doi: 10.1016/j.euo.2024.05.012).
Response: We thank the reviewer for their insightful recommendation. In agreement, we have expanded our discussion to include alternative treatments for BCa following BCG failure (page 17, lines 607-624).
Reviewer 2 Report
Comments and Suggestions for Authors
The authors demonstrated that low-dose eribulin altered the subset of NK cells inside tumors in bladder cancer, which promoted anti-tumor function in this manuscript.
The manuscript has originarity in the point of the role of NK cells. It is interesting to me.
I have several coments and suggestions.
1. In the experiment using human bladder cancer cell lines (Fig.2 and Fig.3), you should try to use more than two cell lines. You used only one cell line, T24.
2. You should show the representative figures of flow cytometry gating the population of NK cells.
3. In future direction, dou you suggest how to use eribulin as one of treatment options for bladder cancer? Bladder instillation in non-muscle invasive bladder cancer? Adjunctive therapy to systemic immunotherapy in advanced bladder cancer?
Author Response
Reviewer 2
In the experiment using human bladder cancer cell lines (Fig.2 and Fig.3), you should try to use more than two cell lines. You used only one cell line, T24.
Response: Thank you for the suggestion. At this time, we’re unable to include additional bladder cancer cell lines in this experiment due to resource constraints. However, we have validated some of our findings in our in vivo model and using our clinical samples. We appreciate your feedback and we have edited our discussion to include that these findings need to be validated in multiple cell lines in future studies (page 17, lines 629-633).
You should show the representative figures of flow cytometry gating the population of NK cells.
Response: We have added a representative figure demonstrating our gating strategy to the supplementary material figure 5.
In future direction, do you suggest how to use eribulin as one of treatment options for bladder cancer? Bladder instillation in non-muscle invasive bladder cancer? Adjunctive therapy to systemic immunotherapy in advanced bladder cancer?
Response: We thank the reviewer for their valuable insights and have incorporated them into our discussion, addressing the suggested points (page 17, lines 624-628).